# Proteomic Exploration of Paraoxonase 1 Function in Health and Disease

**DOI:** 10.3390/ijms24097764

**Published:** 2023-04-24

**Authors:** Hieronim Jakubowski

**Affiliations:** 1Department of Biochemistry and Biotechnology, University of Life Sciences, 60-637 Poznań, Poland; jakubows@rutgers.edu; Tel.: +48-973-972-8733; Fax: +48-973-972-8981; 2Department of Microbiology, Biochemistry and Molecular Genetics, International Center for Public Health, New Jersey Medical School, Rutgers University, Newark, NJ 07103, USA

**Keywords:** paraoxonase 1, homocysteine, proteostasis, oxidative stress, cardiovascular disease, Alzheimer’s disease

## Abstract

High-density lipoprotein (HDL) exhibits cardio- and neuro-protective properties, which are thought to be promoted by paraoxonase 1 (PON1), a hydrolytic enzyme associated with an HDL subfraction also enriched with an anticoagulant protein (PROS1) and amyloid beta-transport protein clusterin (CLU, APOJ). Reduced levels of PON1 activity, characterized biochemically by elevated levels of homocysteine (Hcy)-thiolactone, oxidized lipids, and proteins modified by these metabolites in humans and mice, are associated with pathological abnormalities affecting the cardiovascular system (atherothrombosis) and the central nervous system (cognitive impairment, Alzheimer’s disease). The molecular bases of these abnormalities have been largely unknown. Proteomic and metabolic studies over the past decade have significantly contributed to our understanding of PON1 function and the mechanisms by which PON1 deficiency can lead to disease. Recent studies discussed in this review highlight the involvement of dysregulated proteostasis in the pro-oxidative, pro-atherothrombotic, and pro-amyloidogenic phenotypes associated with low PON1 activity.

## 1. Introduction

Paraoxonase 1 (PON1) is a calcium-dependent hydrolytic enzyme, expressed in the human kidney, liver, colon [1], and brain [2,3], circulating in the blood attached to a subfraction of the high-density lipoprotein (HDL), and present in various organs, including the brain [4]. Proteomic studies have shown that the PON1-containing HDL particles, which represent 5% of total HDL, are enriched in several other important proteins such as PROS1, IGHG1, A2M, ALB, IGLC2, TF, and CLU [5]. The *PON1* gene is located in the *PON* cluster, which also contains the *PON2* and *PON3* genes, on the long arm of chromosome 3 (position 7q21.3), and has several polymorphisms, including the Q192R [6], which involves a change from glutamine (the Q variant) to arginine (the R variant) at position 192 of the amino acid sequence of the PON1 protein and affects its hydrolytic activity with natural [7] and non-natural [6] substrates. Historically, the hydrolytic activity of PON1 has been assayed with non-natural substrates such as the organophosphate paraoxon (for which the PON1 enzyme has been named) and phenyl acetate [6] (Figure 1), suggested to be surrogates for an endogenous substrate that promotes atherogenesis [8]. In addition to participating in HDL-mediated processes [9,10], PON1 participates in sulfur amino acid metabolism by hydrolyzing homocysteine (Hcy)-thiolactone [7,11,12] (Figure 1), a metabolite produced during editing of Hcy by methionyl-tRNA synthetase [13] in protein biosynthesis [14,15].

Recently, Hcy-thiolactone has been shown to predict myocardial infarction in coronary artery disease patients [16] and to promote the accumulation of amyloid beta (Aβ), a hallmark of Alzheimer’s disease (AD), in a mouse model of AD [3]. The involvement of Hcy-thiolactone in disease can be explained by its ability to modify protein lysine residues and impair protein structure/function [15]. Specifically, modification of fibrinogen by Hcy-thiolactone increases the resistance of blood clots to fibrinolysis in vitro [17] and explains the association of Hcy-thiolactone with fibrin clot lysis time recently found in humans in a large randomized controlled trial [18]. Hcy-thiolactone can also accelerate the development of AD by upregulating the expression of amyloid precursor protein (APP) and its amyloidogenic processing, leading to increased generation of Aβ [3]. The PON1 activity with Hcy-thiolactone as a substrate (HTLase) (Figure 1) is strongly positively correlated with the paraoxonase activity in various populations (Figure 2A,B) (the United States [7], Poland [19], the United Kingdom [20], and the Netherlands [21]), but not with the arylesterase activity (measured with phenylacetate as a substrate) (Figure 2C) [21].

PON1 is responsible for essentially all HTLase activity in the blood [11,12], while bleomycin hydrolase (BLMH) [22,23] is a major HTLase in other tissues [15]. Hcy-thiolactone levels are significantly higher in individuals with low HTLase/paraoxonase activity (*PON1-192QQ*) compared with individuals with high HTLase/paraoxonase activity (*PON1-192RR*) [24], suggesting that the paraoxonase activity might reflect the physiological HTLase activity of PON1 (Figure 3A). The arylesterase activity is much less affected by the *PON1-Q192R* variation than the paraoxonase activity [5,24,25,26,27] and appears to reflect the concentration of the PON1 protein [5,25].

The paraoxonase activity affects Hcy-thiolactone levels in humans; lower Hcy-thiolactone levels being observed in individuals with higher paraoxonase activity (Figure 3A). In contrast, Hcy-thiolactone levels are not affected by the arylesterase activity (Figure 3B) or PON1 protein levels (Figure 3C) measured with an anti-PON1 antibody [24].

Enzymological studies showed that paraoxon and phenylacetate are noncompetitive inhibitors of the HTLase activity [11] and that phenyl acetate is a mixed type inhibitor of paraoxonase activity [28], suggesting that different substrates bind either to separate active sites or to different conformations of one active site of PON1 [11]. Crystallographic studies of the engineered rabbit PON1 variants in the free form and in a complex with an inhibitor (2-hydroxyquinoline, a non-hydrolysable lactone analog that inhibits the arylesterase, paraoxonase, and lactonase activities of PON1) support the latter possibility by identifying three distinct conformations of PON1 [29]. Site-directed mutagenesis studies showed disparate effects of some mutations on the paraoxonase, arylestrase, and lactonase activities, suggesting that each of these three activities is catalyzed by a different set of active site residues [29]. These structure/function studies provide the molecular basis for the non-competitive inhibition of the HTLase activity by phenyl acetate and paraoxon [11] and the mixed-type inhibition of the paraoxonase activity by phenyl acetate [28].

In vitro enzymological studies led to the suggestion that the native physiological activity of PON1 is its lactonase activity [30,31]. Some recent studies even state that lactonase activity is “the established native physiological activity of PONs” [32]. Apart from the awkwardness of such pronouncements (how can you establish physiological activity without physiological, i.e., in vivo, studies?), there is no evidence that PON1 can hydrolyze lipophilic lactones in vivo. A possible support for this suggestion is provided by findings that levels of 5,6-DiHET (5,6-dihydroxyeicosatrenoic acid) lactone, a product of spontaneous lactonization of 5,6-epoxyeicosatrienoic acid (5,6-EET), are elevated in the kidneys of *Pon1*^−/−^ mice [33]. However, as 5,6-EET levels were not quantified in these mice, it cannot be excluded that higher 5,6-DiHET lactone levels were reflecting higher 5,6-EET levels in *Pon1*^−/−^ kidneys rather than the effect of the *Pon1* genotype. Further, there was no information in ref. [33] regarding whether *Pon1*^−/−^ and C57BL6 mice used for the 5,6-DiHET lactone assays were from the same litters or whether unrelated C57BL6 mice were used. Thus, the significance of this finding is unclear. A recent study that failed to identify any endogenous lipophilic lactones in human serum [34] does not support a suggestion that PON1 can metabolize lipophilic lactones either.

Proteomic and metabolic studies over the past decade or so have significantly contributed to our understanding of the cardio- and neuro-protective functions of PON1. Recent studies, discussed in the present review, highlight the involvement of dysregulated proteostasis in the pro-oxidative, pro-atherothrombotic, and pro-amyloidogenic phenotypes associated with attenuated PON1 activity.

## 2. PON1, Lipid Oxidation, and CVD

PON1 has been extensively investigated in the context of cardiovascular disease in mouse models and in humans [25]. Initial studies in mouse models have shown that deletion of Pon1 in mice causes increased sensitivity to the toxic effects of organophosphate insecticides (reviewed in [35]) and increased susceptibility to atherosclerosis induced by a high-fat diet [36] or ApoE depletion [37]. However, while the mechanistic role of PON1 in the detoxication of specific organophosphates is clear-cut and well understood [35], its protective role against atherosclerosis is not fully understood, in part due to the undefined nature of its putative pro-atherogenic targets. As oxidative stress is associated with CVD in general [38], it may not be surprising that elevated levels of oxidative stress markers can be found in Pon1-depleted mice and humans with low PON1 activity. However, how low PON1 activity can lead to oxidative stress is not fully understood. Of the numerous studies of PON1 in relation to CVD, some also included analyses of lipid oxidation.

In *Pon1*^−/−^ mice, increased atherosclerosis was accompanied by increased oxidative stress manifested by elevated levels of lipid peroxides in isolated HDL [36], increased oxidized phospholipid epitopes in plasma, bioactive oxidized phospholipids in isolated endogenous intermediate density lipoprotein/LDL, and increased expression of oxidative stress-responsive genes such as heme oxygenase-1, PPARγ, and oxidized LDL-R in the liver [37]. Although there were no lipid hydroperoxides in freshly isolated LDL from any Pon1 genotype, LDL from *Pon1*^−/−^ mice stimulated lipid hydroperoxide generation and monocyte transmigration more than LDL from *Pon1*^+/+^ mice in a coculture model. These findings suggest that LDL from *Pon1*^−/−^ mice is altered to become more susceptible to oxidation. Pretreatment with purified human PON1 inhibited lipid hydroperoxide formation in LDL [36]. Overexpression of human PON1 in *LDL*^−/−^ mice increased PON1 paraoxonase activity 4.4-fold, significantly reduced plaque-associated oxLDL and titers of autoantibodies against malondialdehyde (MDA)-modified LDL, and reduced plaque volume by 80% [39]. Overexpression of human PON1 in *ApoE*^−/−^ mice increased paraoxonase activity in HDL by 60%, stabilized atherosclerotic plaques, and significantly reduced plaque area by 20–30% [40].

The first large-scale prospective study that analyzed the relationship of PON1 activities with systemic oxidative stress and their prognostic value as predictors of future risk for cardiovascular disease (CVD), involving 1339 patients (with CVD n = 1116, 65-year-old, 72% male; without CVD n = 283, 57-year-old, 48% male) undergoing coronary angiography [26], showed that at baseline, low PON1 activity (paraoxonase and arylesterase) and PON1-192QQ genotype were associated with increased levels of oxidative stress markers, such as specific species of oxidized fatty acids (5-, 8-, 9-, 11-, 12-, 15-hydroxyeicosatetraenoic acids (HETEs), 9-, 13-hydroxyoctadecadienoic acids (HODEs), 8-isoprostaneisoprostane prostaglandin F_2α_ (8-isoPGF_2α_)) measured in 150 age-, sex-, and race-matched patients. Participants carrying 192QQ alleles vs. 192QR + 192RR alleles showed elevated risk of major adverse cardiac events (MACE, defined as the sum of death, myocardial infarction (MI), and stroke events) (18.0% vs. 13.6%, adjusted hazard ratio 1.48, *p* = 0.01) and all-cause mortality (11.1% vs. 6.75%, adjusted hazard ratio 2.05, *p* = 0.001) compared with carriers of 192RR and 192QR alleles over the ensuing 3 years but showed no association with nonfatal MI and stroke. In contrast, low PON1 activity predicted a greater incidence of MI and stroke, all-cause mortality, and MACEs. For example, participants with the lowest paraoxonase and arylesterase activities of PON1 (1st quartile) had a greater incidence of MACEs (25.1% and 23.5%, respectively) compared to those with the highest PON1 activities (4th quartile) (7.3% and 7.7%, respectively). The adjusted hazard ratios for nonfatal MI and stroke, all-cause mortality, and MACEs in the highest vs. lowest PON1 activity quartiles were 4.4, 2.4, and 3.4, respectively, for paraoxonase and 4.5, 2.2, and 2.9, respectively, for arylesterase and were independent in multivariate analysis in models (separate for paraoxonase and arylesterase) adjusted for all traditional cardiac risk factors, including the Framingham ATP-III risk score (including diabetes status), log C-reactive protein, body mass index, and medication use (statins and aspirin). This study has provided strong evidence demonstrating that PON1 Q192R polymorphism and PON1 activity influence systemic oxidative stress and predict prospective cardiovascular risk.

Another large-scale prospective study involving 3668 stable subjects without acute coronary syndrome undergoing elective coronary angiography demonstrated serum PON1 activities predict long-term cardiovascular risk [27]. Specifically, low serum arylesterase and paraoxonase activities predicted long-term MACE, independent of traditional clinical and biochemical risk factors, and provided incremental value in reclassifying subjects who are at higher risk of long-term MACE, with arylesterase activity having greater prognostic value (quartile 4 vs. 1: hazard ratio 2.63, *p* < 0.01) than paraoxonase activity (quartile 4 vs. 1: hazard ratio 1.63, *p* < 0.05). Although the distinct single nucleotide polymorphisms (SNPs, including Q192R) within the *PON1* gene, identified by a genome-wide association study, were highly significantly associated with serum paraoxonase (*p* = 1.2 × 10^−303^) and arylesterase (*p* = 5.0 × 10^−116^) activity, these variants were not associated with future risk of MACE in an angiographic cohort (n = 2136), consistent with analyses from the CARDIoGRAM consortium, which did not show any association of these SNPs with history of prevalent CAD or MI in ≈80 000 subjects. However, an earlier study with a smaller cohort (n = 1399) from the same research group, reported an association of Q192R variants with risk of MACE [26]. Observations that arylesterase activity provided prognostic value in both primary prevention (no evidence of CAD) and secondary prevention (evidence of CAD) subjects were interpreted as implying that a lack of antioxidant defense (due to low PON1 activity) may promote greater vulnerability to oxidative stress and to the progression of CAD.

A cross-sectional population-based study of young adults (n = 1895, 32-year-olds, 46% male) examined a relationship between PON1 activity, the rs669 SNP (Q192R), and conjugated dienes in lipoprotein lipids [41], a measure of oxidized lipoprotein lipids [42], which correlated with MDA and hydroperoxides [43]. In multiple regression models, PON1 paraoxonase activity was negatively associated with oxLDL lipids (*p* = 0.0001) but not with oxHDL lipids, and there was also a tendency for an association with oxLDL protein (*p* = 0.08). Stronger activity-oxLDL lipid associations were observed in the RR192 carriers than in the QQ192 carriers. Although the PON1 rs662 SNP was strongly associated with paraoxonase activity, it was not associated with oxLDL lipids or proteins. The finding that PON1 paraoxonase activity is not associated with oxHDL lipids [41] is consistent with another finding showing that increased paraoxonase activity was associated with decreased HDL antioxidant capacity in a small population of young adults [44].

In a case-control study, angiographically confirmed CAD patients (n = 105) had significantly increased levels of plasma 8-isoprostane F2 (8-iso-PGF2α, produced by the non-enzymatic peroxidation of arachidonic acid in membrane phospholipids) and reduced paraoxonase and arylesterase activities of PON1 compared to healthy controls (n = 45) [45]. PON1 paraoxonase and arylesterase activities were significantly associated negatively with the severity of CAD (Gensini score) in univariate analyses, while 8-iso-PGF2α was associated positively. These associations were also observed in multiple regression analyses in models adjusted for age, sex, smoking, hypertension, diabetes, statins, HDL cholesterol, and triglycerides. These associations suggest that PON1 may protect phospholipids from oxidative damage [45].

Another large-scale study investigated the association of PON1 with cardiovascular risk in healthy individuals (48-year-olds, 52% women) without a known history of CVD at baseline [46]. The study prospectively measured PON1 arylesterase activity in 6902 study participants, who were followed up for 9.3 years, during which 730 adverse cardiovascular events occurred. In Cox regression analyses, PON1 arylesterase was weakly associated with HDL cholesterol and APO-A1, and there was a log-linear negative association with CVD risk, dependent in part on HDL cholesterol. HDL cholesterol was strongly associated with CVD risk in models with or without PON1 arylesterase. A meta-analysis of six studies with 15,064 individuals and 2958 incident MACEs led to the conclusion that, although there was a negative relationship between PON1 and MACEs, PON1 does not add value to cardiovascular risk stratification beyond conventional cardiovascular risk factors [46].

A recent meta-analysis compared PON-1 arylesterase plasma/serum levels between CAD and non-CAD patients in studies published between 1 January 2000 and 1 March 2021 [47]. A total of 20 studies were selected that met the inclusion criteria, and a total of 5 417 patients (3 364 with CAD and 2 053 without CAD) were included in the analysis. The meta-analysis showed that PON1 arylesterase activity was significantly lower in CAD patients compared to non-CAD controls. In CAD patients with diabetes mellitus, the PON-1 arylesterase activity was significantly lower compared with CAD patients without diabetes [47].

The PON1 genotype has also been linked to abdominal aortic aneurism (AAA). Specifically, a study with 423 AAA patients and 423 matched controls found that the PON1 haplotype consisting of Leu55, Arg192, and Trp194 differed in frequency between control subjects (0.374) and AAA patients (0.288) (*p* < 0.042), suggesting a protective effect of this haplotype against AAA [48]. Other studies found that reactive oxygen species (ROS) and reactive nitrogen species (RNS) were associated with AAA formation in animal models and in humans [49], suggesting that PON1 may protect against aneurysms through its anti-oxidative function [50].

A large-scale prospective study analyzed relationships between PON1 rs705379 (-108C > T), rs854560 (L55M), and rs662 (Q192R) SNPs and outcomes in uremic patients on hemodialysis (n = 1407, 67-year-olds, 56% male, 0.01–34 years of renal replacement therapy (RRT)) and their prognostic value as predictors of CVD risk [51]. The PON1 rs854560 allele T was associated with dyslipidemia (*p* = 0.014) and increased risk of ischemic cerebral stroke (OR 1.38, 1.02–1.85, *p*  =  0.034), while the PON1 rs705379 TT variant was associated with CVD mortality (HR 1.27, 95% CI 1.03–1.57, *p*  =  0.025), while PON1 rs662 was not.

Another study examined the effects of cigarette smoking on the associations between PON1 SNPs rs705379, rs854560, and rs662 and cardiovascular mortality in hemodialysis (HD) patients [52]. Cardiovascular, cardiac, and coronary heart disease (CHD)- and non-CHD-related deaths were analyzed in HD cigarette smokers (n = 82, 57-year-olds, 85% male, 46% on RRT, 6.8 (0.7to 28) years of RRT, and age at death, 63.5 (31 to 86) years) and HD non-smokers (n = 239, 68-year-olds, 38% male, 46% on RRT, and 5.7 [0.3–29] years of RRT, age at death, 63.5 (31 to 86) years). Smokers carrying the PON1 promoter rs705379 TT variant were at significantly higher risk for CVD and total mortality. Non-diabetic smokers also showed a significantly increased risk of mortality. In diabetic non-smokers, the rs705379 T SNP was associated with CHD-related mortality, while the rs854560 T SNP was associated with lower CVD mortality in non-diabetic smokers (*p* = 0.008). In diabetic smokers, the rs662 G SNP was associated with higher cardiac mortality (*p* = 0.005). In all non-smokers and non-diabetic non-smokers, the rs662 G was associated with CVD mortality (*p* = 0.020 and *p* = 0.018, respectively). These findings suggest that genotyping *PON1* SNPs might help to identify uremic patients at higher risk of mortality [52].

To identify genetic determinants of serum paraoxonase and arylesterase activities, genome-wide association studies (GWAS) have been carried out. These studies, involving 2136 Caucasian subjects, identified a major locus on chromosome 7, containing the *PON1* gene, that controls these activities [27]. There were no other loci in the genome that were significantly associated with PON1 activity. The lead SNP (rs2057681) for serum paraoxonase activity at this locus is in linkage disequilibrium with SNP rs662 (Q192R), an amino acid substitution in PON1 responsible for a significant increase of paraoxonase activity. In contrast to paraoxonase activity, these two SNPs are associated with a significant decrease in arylesterase activity. The lead SNP for serum arylesterase activity (rs854572) is in the promoter region of the PON1 gene and is also associated with increased paraoxonase activity. Interestingly, the four lead SNPs for arylesterase activity (rs854570, rs854572—promoter -108, rs705382, rs757158) differ from the lead SNPs for paraoxonase activity (rs2269829, rs662—Q192R, rs2057681, rs854560—L55M). However, the lead variants affecting paraoxonase and arylestrase activity were not associated with incident or prevalent cardiovascular disease [27].

## 3. Mechanistic Bases of PON1 Involvement in CVD

### 3.1. PON1 Controls NO Synthesis

In addition to promoting reverse cholesterol transport, normal HDL has antioxidative and anti-inflammatory properties [9] and directly affects the vascular endothelium via activation of NO synthesis by eNOS [53,54], thereby promoting endothelial repair [55]. These processes contribute to the protective function of HDL in the cardio-vasculature and are thought to be mediated to a significant extent by PON1 [56], which is carried on a relatively small HDL subfraction, representing 5% of total HDL [5]. Patients with stable CAD or an acute coronary syndrome carry dysfunctional HDL_CAD_, which—in contrast to HDL from healthy subjects—fails to induce endothelial NO synthesis, endothelial anti-inflammatory effects, and endothelial repair [56]. This occurs because HDL_CAD_ activates LOX-1, an endothelial lectin-like oxLDL receptor 1, thereby inducing endothelial PKCβII activation, which in turn inhibits eNOS-activating pathways and eNOS-dependent NO synthesis. These newly acquired properties were due to increased formation of MDA, a product of lipid peroxidation, which impaired PON1 activity and led to the generation of dysfunctional HDL_CAD_ with PKCβII-activating properties and devoid of the antioxidative and anti-inflammatory effects of normal HDL. Modification with MDA of normal HDL impaired its capacity to stimulate endothelial NO production, and this was at least in part mediated through the endothelial LOX-1. Further, HDL isolated from *Pon1*^−/−^ mice did not stimulate endothelial NO production, while supplementation with purified PON1 of HDL isolated from CAD patients or *Pon1*^−/−^ mice partially improved the ability of HDL to stimulate endothelial NO production. These findings show that HDL-associated PON1 activity plays an important role in maintaining the endothelial-atheroprotective effects of HDL, i.e., the capacity to stimulate endothelial NO production. Dysregulation of this fundamental function of PON1/HDL by oxidative stress can account for the increased risk of MACEs in CAD patients.

### 3.2. PON1 Is Not a Redox Protein

It has often been stated that PON1 promotes antioxidant and atheroprotective effects due to its ability to hydrolyze oxidized lipids, e.g., refs [26,57]. The notion that PON1 has the ability to hydrolyze oxidized lipids originated from a study that reported an inhibitory effect of purified PON1 on copper-induced oxidation of LDL, quantified as lipoperoxides and TBARS in an in vitro assay [58]. This assay has been used in other studies of PON1 function [59,60,61,62]. 

Apart from the incorrect inference that a hydrolytic activity is a redox function, there is no convincing evidence that PON1 possesses an intrinsic redox activity. The paraoxonase and arylesterase hydrolytic activities that are routinely quantified in clinical studies cannot be mechanistically linked to red-ox processes. Further, the *PON1-Q192R* genetic polymorphism, which is associated with indices of oxidative stress in vivo [26], has opposite effects on paraoxonase and arylesterase activities: the 192Q allele is associated with low paraoxonase and high arylesterase activity, while the 192R allele is associated with high paraoxonase and low arylesterase activity [24,27,63,64]. Although the initial study reported that a purified PON1 protein preparation had the ability to inhibit copper-induced LDL oxidation and the generation of lipid peroxides [58], this report is controversial. In fact, other studies have clearly demonstrated that PON1 has no intrinsic ability to protect LDL from oxidation and that the putative antioxidant activity was due to contaminants in the purified PON1 preparations [65,66,67]. For example, it has been shown that trace amounts of platelet-activating factor acetyl hydrolase, a serine esterase carried on HDL, contaminate PON1 preparations. PON1 free of PAF acetyl hydrolase contamination (confirmed by the absence of phospholipase activity towards PAF) did not have the ability to detoxify oxidized phospholipids [65]. In assays with copper or the free radical generator 2,2′-azobis-2-amidinopropane, the putative antioxidant activity of PON1 preparations was associated with the detergent present in these preparations but not with arylesterase, lactonase, or phospholipase A2 activities [66]. Finally, copper-induced oxidation of LDL could not be prevented by recombinant PON1 expressed and purified from insect cells harboring a baculovirus vector containing the human PON1 gene [31]. During purification, there was no antioxidant activity co-purifying with PON1 in any of the preparations, and the putative antioxidant activity was associated with a low-mass contaminant and the detergent in the preparation. These findings raise doubts regarding the veracity of the results obtained by examining the effects of PON1 protein preparations on the copper-LDL oxidation assay.

A variation of the copper-LDL oxidation assay, in which conjugated dienes are monitored by absorbance at 234 nm, was used to study the relationship of *PON1* SNPs and arylesterase activity with the oxidation susceptibility of LDL isolated from male individuals with CAD and healthy controls [68]. No active PON1 was present in the LDL oxidation assay because HDL was removed during the preparation of LDL from each individual. This study involved 205 CAD patients (age 70-years-old, >80% stenosis) and 232 controls (age 66-years-old, <15% stenosis), and found that LDL oxidation susceptibility, although strongly correlated with CAD, was not correlated with PON1 arylesterase activity. However, several *PON1* SNPs, including the functional PON1 promoter SNP, *PON1-108C/T*, appeared to be associated with LDL oxidation susceptibility. Although this SNP affects the expression of PON1 and thus arylesterase activity, aryleaterase was not correlated with LDL oxidation susceptibility. The lack of congruency in the relationships of LDL oxidation susceptibility with CAD, PON1 arylesterase activity, and the *PON1-108C/T* SNP raises doubts regarding the adequacy of the approaches used in these studies.

### 3.3. PON1 Interacts with Redox-Related Proteins

Recent studies of proteomes in mice and humans with genetically attenuated PON1 activity have provided new insights into PON1 function. Specifically, these studies suggest that oxidative stress associated with PON1 depletion is caused by disruption of interactions between PON1 and rex-ox-related proteins.

Plasma proteomes, analyzed using label-free mass spectrometry, were examined in healthy participants (n = 100, 49-year-olds, 50% women) recruited from the Poznań, Poland population and in 4-month-old *Pon1*^−/−^ mice (n = 17) and their *Pon1*^+/+^ siblings (n = 8) (50% female) [64]. *PON1-Q192R* polymorphism and *Pon1*^−/−^ genotype induced similar changes in the plasma proteomes of humans and mice, respectively. Pon1 depletion in mice and *PON1-Q192R* polymorphism in humans led to dysregulation of proteins involved in the maintenance of redox homeostasis (Table 1). Redox proteins affected by the *PON1-Q192R* polymorphism and *Pon1*^−/−^ genotype represented 19% (4 out of 21) and 14% (7 out of 50) of total differentiating proteins in humans (Figure 4A) and mice (Figure 4B), respectively. Some of those proteins (Alb and APOM) as well as the complement/coagulation protein clusterin (Clu, ApoJ) are components of HDL [5]. Of these, Clu (ApoJ) is known to be involved in the transport of Aβ from plasma to the brain [69].

In addition to oxidative stress, proteins involved in complement/blood coagulation, lipoprotein/lipid metabolism, immune response, and other processes were dysregulated by PON1 depletion in humans and mice. Bioinformatics analysis using the Ingenuity Pathway Analysis (IPA) resources identified the top molecular network, the “Lipid Metabolism, Molecular Transport, Small Molecule Biochemistry” network, which is affected by low PON1 activity in humans (Figure 5A) and mice (Figure 6A). Proteins in this network participate in lipid metabolism and acute phase/immune response and show strong interactions centering on the lipoproteins LDL and PON1/HDL, the cytokine IL6, and TGFB1. Notably, this network contains redox proteins such as Alb, Blvrd, Ambp, Hpx, Hp, ApoD, and ApoM in mice. Other *PON1*-dependent proteomic changes affected different biological networks in humans and mice: the “Cardiovascular, Neurological Disease, Organismal Injury/Abnormalities” in *PON1-192QQ* vs. *PON1-192QR* and *PON1-192RR* in humans (containing redox-related proteins GPX3, APOD, APOM) (Figure 5B) and the “Humoral Immune Response, Inflammatory Response, Protein Synthesis” (containing redox-related protein Blvrd) (Figure 6B). These findings suggest that PON1 interacts with molecular pathways involved in lipoprotein metabolism, acute/inflammatory response, and complement/blood coagulation, which are essential for blood homeostasis. Dysregulation of these interactions by low PON1 activity can account for its association with cardiovascular and neurological diseases.

### 3.4. HHcy Diet Exacerbates Pro-Oxidative and Pro-Atherogenic Changes in Mouse Proteome

Proteomic studies using label-free mass spectrometry and IPA bioinformatics resources examined plasma proteomes in *Pon1*^−/−^ mice (n = 32) and their *Pon1*^+/+^ siblings (n = 15) fed with a hyper-homocysteinemic (HHcy) diet [70]. Pon1 depletion led to dysregulation of proteins involved in the maintenance of redox homeostasis (Table 2). Redox proteins affected by *Pon1*^−/−^ genotype represented 20% (18 out of 89) and 14% (7 out of 50) of total differentiating proteins in mice fed with the HHcy diet (Figure 4C) and control diet mice (Figure 4B), respectively. Most of the redox proteins affected by the *Pon1^−/−^* genotype in mice fed with a control diet (3 out of 4, 75%) were also affected in HHcy mice, while most redox proteins that were affected by Pon1 depletion in HHcy mice (15 out of 18, 83%) were not affected by Pon1 depletion in control diet animals. These findings clearly show that the metabolic stress of HHcy greatly amplifies the pro-oxidant effects of the *Pon1^−/−^* genotype.

In addition to oxidative stress, proteins involved in processes such as acute phase response, complement/blood coagulation, lipoprotein/lipid metabolism, immune response, purine metabolism, and glucose metabolism were dysregulated by Pon1 depletion in HHcy mice (Table 2, Figure 4C). Some of those proteins (Alb, Clu, A2m, and Pros1) are components of HDL, enriched in the PON1-containing subfraction [5]. These findings suggest that Pon1 interacts with proteins involved in antioxidant defenses and other processes linked to cardiovascular disease. Dysregulation of these processes provides an explanation for the pro-oxidant and pro-atherogenic phenotypes observed in mice and humans with attenuated PON1 levels.

Bioinformatic analyses using IPA resources showed that several redox-related canonical pathways were dysregulated by Pon1 depletion. These included the Iron Homeostasis System pathway, identified both in HHcy diet mice as well as in control diet animals (indicated by a red arrow in Figure 7). In HHcy diet mice, other redox-related canonical pathways were dysregulated by Pon1 depletion, such as glutathione biosynthesis, NRF2-mediated oxidative stress response, thioredoxin, and superoxide radicals degradation pathways (indicated by red arrows in Figure 7). In control diet mice, another redox-related canonical pathway was dysregulated by Pon1 depletion: the heme degradation pathway (indicated by a red arrow in Figure 7).

Three top molecular networks of proteins affected by Pon1 depletion were identified: the “Cardiovascular Disease, Organismal Injury and Abnormalities, Protein Synthesis” network which contains proteins participating in iron metabolism/oxidative stress response (Cp, HP, HPX, and CGLM) that show strong interactions centering on HDL (Figure 8A); the “Hematological Disease, Humoral Immune Response, Inflammatory Response” network that contains oxidative stress response proteins (Prdx2, Prdx6, Park7, and Ppia) that show strong interaction centering on Akt and immunoglobulins (Figure 8B); and the “Cancer, Cellular Compromise, Inflammatory Response” network containing redox-related proteins (Cat, Ctsb, Grn, Gsn, Igfbp3, Pebp1, Serpina3, and Txn) that show interactions centering on the transcription factors Ap1, Tgfβ, and Nfκβ (Figure 8C).

Nineteen proteins that are involved in maintaining redox homeostasis, including Parkinson disease protein 7 (Park7, DJ-1), peroxiredoxin-2 (Prdx2), peroxiredoxin-6 (Prdx6), and thioredoxin (Txn), were significantly downregulated in HHcy Pon1-depleted mice (Table 1). As these proteins protect cells from oxidative stress by detoxifying reactive oxygen species, attenuation of their levels would lead to increased oxidative stress. Impairment of antioxidant defenses due to changes in the expression of red-ox proteins could explain elevated oxidative stress observed in Pon1-depleted mice [36] as well as in humans with reduced PON1 activity [26].

## 4. PON1, Lipid Oxidation, and Alzheimer’s Disease

Accumulating evidence suggests that there is a strong association of CVD and its risk factors with cognitive decline and Alzheimer’s disease (AD) [71]. For example, individuals with subclinical CVD are at higher risk for dementia and AD. Several cardiovascular risk factors exist, such as hypertension, dyslipidemia, and diabetes. Moderate alcohol consumption appears to be protective for both CVD and dementia. However, markers of inflammation predict cardiovascular risk but not dementia [72].

As discussed in the previous section, PON1 is linked to CVD. Thus, it may not be surprising that PON1 is also linked to AD [10,73]. Indeed, PON1 activity is lower in AD and dementia patients compared to healthy controls [74,75,76,77] and correlates with the severity of AD-related cognitive decline [78]. In patients with mild cognitive impairment, PON1 activity predicted global cognition, verbal episodic memory, and attention/processing speed [79]. In mice, *ApoE*^−/−^*Pon1*^−/−^ animals, which have severe carotid atherosclerosis [37], showed AD markers and impaired vasculature in their brains at 14 months, although it was not clear whether brain pathology was caused by *ApoE*^−/−^, *Pon1*^−/−^, or both knockouts [80]. In a mouse model of AD (Tg2576), immunohistochemical fluorescence signals for Pon1 protein in various regions of the brain were found to surround Aβ plaques but could not be colocalized with any brain cell type [81]. However, only a handful of studies examining PON1 in relation to AD included analyses of lipid oxidation.

To identify SNPs associated with a risk of AD, a GWAS of AD patients (n = 756) and ethnically matched control subjects (n = 736) from memory referral clinics in Canada and the United Kingdom was performed [82]. These studies identified SNPs within 26 candidate genes with reported associations with AD, including a SNP in the *PON1* gene (rs2299261). Another study, analyzing frequencies of rs854560 (L55M) and rs662 (Q192R) SNPs in the brains of AD patients (n = 636) and non-demented controls (n = 430) from the Douglas Hospital Brain Bank, Quebec, Canada, found sex-specific effects of these SNPs on AD risk, age of onset, and survival [83]. The L55M polymorphism increased the risk of AD in only men, while M55 and Q292 homozygotes showed increased survival (by 2.5 years) and a later age of onset (by 1.5 years) for both genders. These variants were associated in a sex-specific manner with Aβ levels and senile plaque burden.

A region containing the *PON1* promoter and three first exons was also associated with the risk of AD in a large Caucasian and African American population. This region was defined by four SNPs: rs705381 (–161 C/T), rs705379 (107G/A), rs854565 (intron 1), and rs854650 (L55M) [84]. Two SNPs from this region were associated with a risk of AD in a French Caucasian cohort of AD patients (n = 629) and controls (n = 669) [85]. Another SNP from this region, PON1 T-107C, was more prevalent in AD patients than controls [75].

Meta-analysis of fifteen studies (involving five polymorphisms) on the relationship between PON1 SNPs and AD [86] found that the A allele of the rs705379 SNP (PON1 107/108 G/A) conferred an increased susceptibility to AD (odds ratio 0.7, *p* = 0.002) while the GG genotype decreased AD risk (odds ratio 1.21, *p* = 0.009) in the Caucasian population. This meta-analysis also included the PON2 S311C polymorphism and found that the homozygotic SS311 allele carriers had a decreased risk of AD (odds ratio 0.82, *p* = 0.04). However, 3 other *PON1* polymorphisms (L55M, Q192R, and -161C/T of the PON1 gene) were not associated with AD [86].

As discussed above, in case-control studies, low PON1 activity was associated with AD. In a few studies that also monitored oxidative stress markers, in addition to PON1 activity, increased oxidative stress in AD patients was reported. For example, oxLDL, paraoxonase, arylesterase, and lactonase activities of PON1 and PAF-AH activity were quantified in serum from AD patients (n = 49, 74-year-olds, 41% male, MMSE score = 21 ± 5) and age- and sex-matched control individuals (n = 34) [87] as well as cognition (MMSE score), PON1 activities were significantly reduced (by 22–29%) and oxLDL was significantly increased (by 28%) in AD patients vs. controls; PAF-AH was also elevated (by 45%). There was a significant negative correlation between PON1 activity, but not PAF-AH activity, and oxLDL in both AD patients and controls. PON1 activities and oxLDL levels were also associated with the severity of AD (determined by neuropsychological examination using the MMSE test, a measure of global cognition). Specifically, patients with moderate (MMSE score = 11–24) and severe AD (MMSE score < 10) showed lower PON1 activities and higher oxLDL compared with patients with mild AD (MMSE score > 24). Although hydrolysis of oxidized phospholipids by PAF-AH generates lysophospholipids (lysophosphatidylcholine) and oxidized free fatty acids, which have potent biological activity, PAF-AH activity was not associated with oxLDL nor the severity of AD. These findings suggest a role for PON1 in oxLDL metabolism in AD.

Another study evaluated LDL oxidation in late-onset AD patients (n = 54, 77-year-old, 19% male, MMSE score = 18) and healthy elderly subjects (n = 51, 77-year-old, 27% male, MMSE score = 29) and its interaction with *PON1-107C/T* and apolipoprotein E (*APOE*) genotypes [88]. AD patients and controls with the *PON1-107TT* genotype showed significantly increased plasma levels of oxLDL compared with the *PON1-107CC/CT* genotype. The distribution of lipoprotein cholesterol in AD patients shifted toward a greater prevalence of smaller, denser LDL. There was a significant association between levels of smaller, denser LDL and oxLDL in AD patients. The *APOE* genotype did not modulate lipoprotein distribution. The authors concluded that their results suggest that the association between *PON1-107TT* polymorphism and AD could be mediated by increased oxidation of plasma LDL.

Relationships between PON1, lipid peroxidation, and dementia were examined in patients with AD (n = 63), vascular dementia (n = 40), and mixed dementia (n = 33) [78]. Malondialdehyde/thiobarbituric acid reactive substances were increased more in vascular dementia than in AD. In patients with vascular involvement, increases in MDA/TBARS reflected the extent of global cortical atrophy. PON1 arylesterase activity was decreased in patients with dementia, more so in those with severe cognitive deficits. In patients with vascular dementia, a reduction in PON1 arylesterase activity reflected the increase in medial temporal lobe atrophy and brain ischemia. The authors concluded that PON1 activity and MDA/TBARS markers of oxidative stress were associated with vascular dementia and brain atrophy rather than cognitive decline.

### 4.1. PON1 and Cognition

The prevalence of mild cognitive impairment (MCI), “a cognitive decline greater than that expected for an individual’s age and education level but that does not interfere notably with activities of daily life” [89], is ~16% in individuals over 70 [90,91]. Half the MCI cases convert to AD within 5 years after diagnosis [89]. Low PON1 arylesterase activity, found in individuals with MCI [78,92], has been associated with an increased risk of developing vascular dementia [93].

A recent study examined relationships between PON1 status and functional and structural aspects of brain function [79]. In that study, PON1 activity and genotype were assessed as predictors of cognition and brain atrophy in individuals with MCI (78-year-olds, n = 196, 60% women) who had an MMSE score > 24/30 and no evidence of dementia and were subjected to B vitamin (n = 95; daily oral supplement tablets with folic acid, vitamin B_12_, and B_6)_ or placebo (n = 101) treatment for two years. Cognition was analyzed by neuropsychological tests. Brain atrophy was quantified by MRI.

In Pearson analysis at baseline, PON1 arylesterase activity was associated with over a dozen variables, including measures of cognition in the semantic memory, verbal episodic memory, and attention/processing speed domains. PON1 paraoxonase activity was associated with four variables, and only two of these variables were also associated with arylesterase (the *PON1 Q192R* genotype and silicon). However, in multiple regression analysis, baseline PON1 arylesterase activity remained associated only with iron, creatinine, and the *PON1 Q192R* genotype. In contrast, baseline paraoxonase activity remained associated only with the *PON1 Q192R* genotype [79].

Multiple regression analysis showed that, in the placebo group, baseline PON1 arylesterase activity (but not paraoxonase activity or the *PON1 Q192R* genotype) was a significant negative predictor of global cognition, verbal episodic memory, and attention/processing speed at the end of the study two years later (Table 3). Although brain atrophy rate was negatively associated with global cognition and attention/processing speed, baseline PON1 arylesterase activity was not associated with brain atrophy rate. These findings suggest that PON1 affects functional but not structural aspects of cognition [79].

In addition to PON1 arylesterase, other factors predicted cognition (Table 3). Specifically, baseline iron [94] and triglycerides predicted global cognition; baseline fatty acids predicted attention/processing speed; baseline anti-*N*-Hcy-protein autoantibodies predicted global memory and attention/processing speed [95], while the *BDNF V66M* genotype predicted verbal episodic memory at the end of the study. B vitamin treatment abrogated associations of PON1 and other variables with cognition (Table 4). Taken together, these findings identify PON1 as a new factor associated with impaired cognition in MCI [79].

### 4.2. Mechanistic Bases of PON1 Involvement in AD

Ox-LDL is known to be cytotoxic, pro-inflammatory, and induces oxidative stress in brain cells such as astrocytes, microglia, and neurons [53,54]. Notably, Aβ binds to ox-LDL and accelerates macrophage foam cell formation [55]. These findings suggest that ox-LDL can be directly involved in the development of AD.

PON1 is important to the detoxification of neurotoxic agents such as organophosphate pesticides, which are potent inhibitors of acetylcholinesterase; low PON1 activity increases susceptibility to these agents [36]. Occupational exposure to organophosphates is known to increase the risk of developing AD later in life [96,97]. Classical studies have shown that treatments with doses of chlorpyrifos oxone that do not affect *Pon1*^+/+^ mice induce convulsions and death in *Pon1*^−/−^ animals [36]. Pon1 is also important for the detoxification of another neurotoxic agent, Hcy-thiolactone: treatments with Hcy-thiolactone induce seizures with significantly higher frequency and lower latency in *Pon1*^−/−^ mice compared to *Pon1*^+/+^ animals [12].

#### 4.2.1. Pon1 Depletion Induces Pro-Neurodegenerative Changes in Mouse Brain Proteome

Proteomic studies of *Pon1*^−/−^ vs. *Pon1*^+/+^ mice show that, in addition to controlling Hcy-thiolactone and *N*-Hcy-protein levels [12], Pon1 is important for maintaining cellular proteostasis [98]. Pon1 was found to interact with diverse sets of cellular proteins in an organ-specific manner, and those interactions were modulated by HHcy. In *Pon1*^−/−^ mouse brains, levels of proteins participating in antioxidant defenses (Sod1, DJ-1), brain-specific function (Nrgn), and cytoskeleton assembly (Tbcb) are significantly reduced, while the cytoskeleton assembly protein CapZa2 is increased, relative to *Pon1*^+/+^ animals [98]. In the brains of HHcy *Pon1*^−/−^ mice, proteins involved in antioxidant defenses (Prdx2, DJ-1), brain-specific function (Ncald, Nrgn, Stmn1), energy metabolism (Ak1), cell cycle (GDI1, Ran), cytoskeleton assembly (Tbcb), and unknown function (Hdhd2) are upregulated (Table 5). Of particular interest to the present discussion of the relationships between PON1, oxidative stress, and AD are findings showing that Pon1 depletion affects the expression of Sod1, Prdx2, and DJ-1 proteins that are involved in oxidative stress and are associated with AD [98].

Clusterin (CLU or APOJ), involved in the transport of amyloid beta (Aβ) from plasma to the brain in humans (reviewed in [69]), is carried on a distinct HDL subspecies, representing 5% of total HDL, that contains three major proteins: PON1, CLU, and APOA1 [99]. Notably, levels of Clu (ApoJ) are significantly elevated in the plasma of *Pon1*^−/−^ vs. *Pon1*^+/+^ mice both in the absence [64] and presence of HHcy [70] (Table 2).

#### 4.2.2. Pon1 Depletion Induces Accumulation of Amyloid β in Mouse Brain

The involvement of PON1 in the development of AD was studied in a new mouse model of AD, the *Pon1*^−/−^5xFAD mouse [3]. 5xFAD mice overexpress the K670N/M671L (Swedish), I716V (Florida), and V717I (London) mutations in human APP(695), and the M146L and L286V mutations in human PS1, and accumulate high levels of Aβ42 beginning around 2 months of age [100]. Dysregulated mTOR signaling and autophagy have been linked to Aβ accumulation in AD [101,102], and H4K20me1 demethylation by PHF8 is important for maintaining the homeostasis of mTOR signaling [103]. Thus, the study [3] examined how Pon1 depletion affects mTOR signaling, autophagy, and Aβ accumulation. To elucidate the mechanism involved, these processes were further studied in mouse neuroblastoma cells harboring N2a-APP_swe_ cells, which overproduce Aβ.

The study found that Pon1 protects from amyloidogenic APP processing to Aβ in the mouse brain (Figure 9) and identified the mechanistic basis of the protective role of Pon1 in the central nervous system. Specifically, Pon1 depletion significantly downregulated histone demethylase Phf8 and upregulated the H4K20me1 histone mark. mTOR, phospho-mTOR, and App were upregulated while autophagy markers Bcln1, Atg5, and Atg7 were downregulated at protein and mRNA levels in the brains of *Pon1*^−/−^5xFAD vs. *Pon1*^+/+^5xFAD mice. Pon1 depletion in N2a-APP_swe_ cells by transfection with siRNA targeting the *Pon1* gene downregulated Phf8 and upregulated mTOR due to increased H4K20me1 binding to the *mTOR* promoter. Upregulation of mTOR signaling inhibited autophagy and significantly increased APP and Aβ levels. Treatments with Hcy-thiolactone or *N*-Hcy-protein (metabolites that were previously found to be elevated in *Pon1*^−/−^ mice [12]) or Phf8 depletion by siRNA interference similarly elevated Aβ levels in N2a-APP_swe_ cells [3]. Notably, Phf8 depletion did not affect App expression in these cells, indicating that upregulation of Aβ was not caused by App overexpression [104].

Pon1 depletion caused changes in the Phf8->H4K20me1->mTOR->autophagy pathway (Figure 9) in the mouse brain akin to the changes induced by HHcy [3], suggesting involvement of the same Hcy metabolites. Indeed, previous work demonstrated that a common primary biochemical outcome of Pon1 depletion or of HHcy was essentially the same: Pon1 depletion [12,21] or HHcy [105] led to elevations of Hcy-thiolactone and *N*-Hcy-protein levels. Taken together, these findings provide evidence that increased accumulation of Aβ in Pon1-depleted brain is mediated by effects of Hcy metabolites on mTOR signaling and autophagy [3], which could explain the associations of low PON1 activity [10] as well as HHcy [106] with AD.

## 5. Conclusions

Proteomic analyses generated new insights into Pon1 function by identifying proteins and molecular pathways affected by Pon1 deficiency. The accumulating evidence discussed above strongly suggests that Pon1 depletion dysregulates cellular and extracellular proteostasis by impairing epigenetic regulation, upregulating mTOR signaling, and inhibiting autophagy, which leads to increased generation of amyloid β. Pon1 depletion also induces pro-oxidative, pro-inflammatory, and pro-atherogenic changes in plasma and tissue proteomes. These changes in gene expression are exacerbated by HHcy and are like the changes induced by Hcy-thiolactone and N-Hcy-protein. These processes can directly or indirectly lead to cardiovascular disease and worse outcomes in CVD patients, as well as to neurological impairments associated with dementia and Alzheimer’s disease.

## Figures and Tables

**Figure 1 ijms-24-07764-f001:**
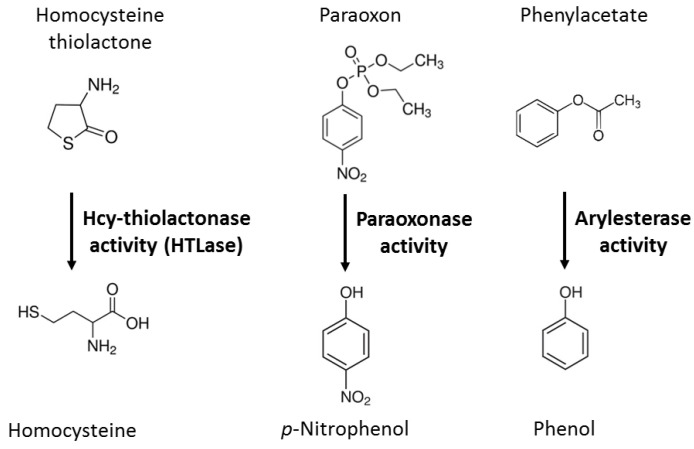
Enzymatic activities of PON1. Hcy-thiolactone is a naturally occurring substrate of PON1.

**Figure 2 ijms-24-07764-f002:**
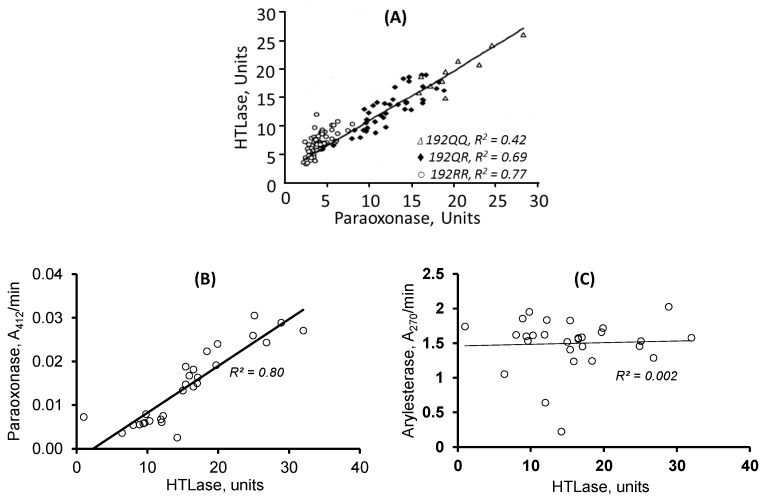
HTLase activity of PON1 is positively correlated with paraoxonase but not with arylesterase activity. Enzymatic activities of serum PON1 were quantified in individuals from Poland (**A**) and the Netherlands (**B**,**C**). Adapted from ref. [19,21].

**Figure 3 ijms-24-07764-f003:**
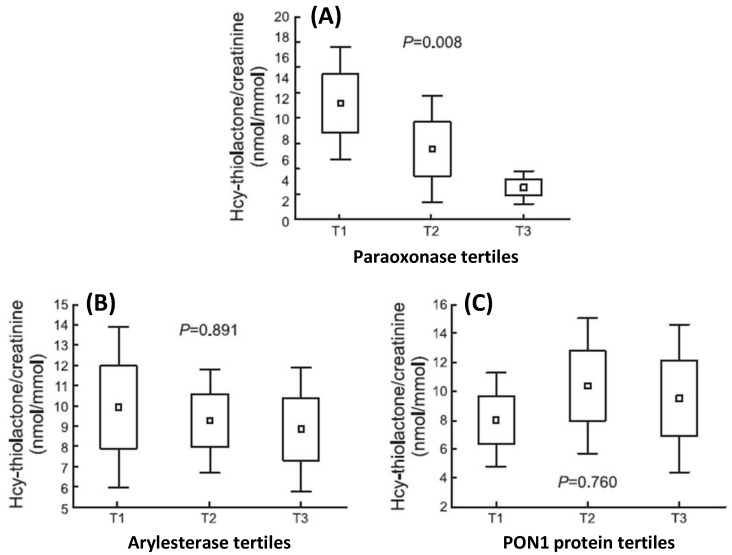
Hcy-thiolactone levels in humans are affected by the paraoxonase activity of PON1 (**A**), but not by the arylesterase activity (**B**) or PON1 protein levels (**C**). T1, T2, and T3 denote the lowest, middle. and highest tertiles, respectively. Adapted from ref. [24].

**Figure 4 ijms-24-07764-f004:**
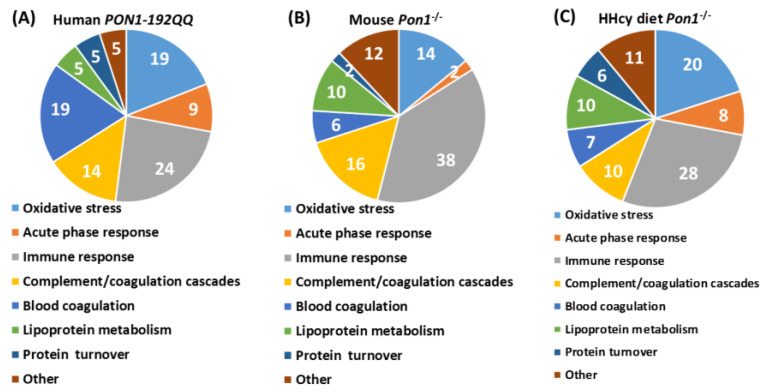
Relative numbers of proteins (5) involved in indicated molecular processes affected by the *PON1-192QQ* genotype in humans (**A**) and by the *Pon1*^−/−^ genotype in mice fed with a control (**B**) or HHcy diet (**C**). Recalculated from data in refs [64,70].

**Figure 5 ijms-24-07764-f005:**
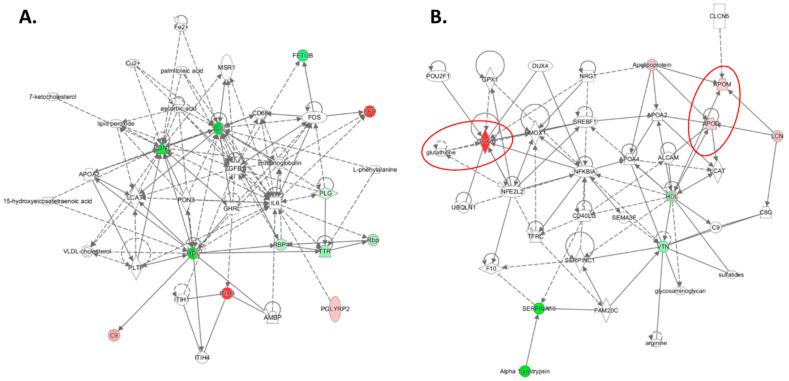
Molecular networks associated with *PON1-Q192R* polymorphism in humans. (**A**). Lipid metabolism, molecular transport, and small molecule biochemistry. (**B**). Cardiovascular disease, neurological disease, organismal injury, and abnormalities. This network contains redox-related proteins (GPX3, APOD, and APOM). Red ovals highlight redox-related proteins affected by the *PON1-Q192R* polymorphism. Adapted from ref. [64].

**Figure 6 ijms-24-07764-f006:**
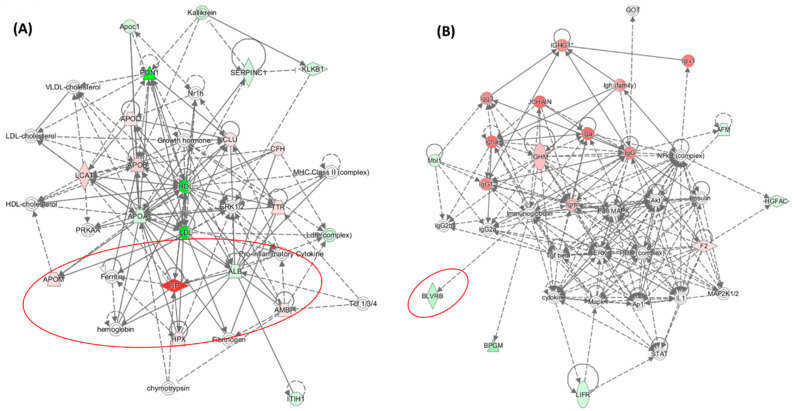
Top molecular networks associated with the *Pon1*^−/−^ genotype in mice. (**A**) Lipid metabolism, molecular transport, and small molecule biochemistry. This network contains redox-related proteins Alb, Ambp, Apom, Hp, and Hpx. (**B**) Humoral immune response, inflammatory response, and protein synthesis. This network contains the redox-related protein BLVRD. Red ovals highlight redox-related proteins dysregulated by Pon1 depletion. Adapted from ref. [64].

**Figure 7 ijms-24-07764-f007:**
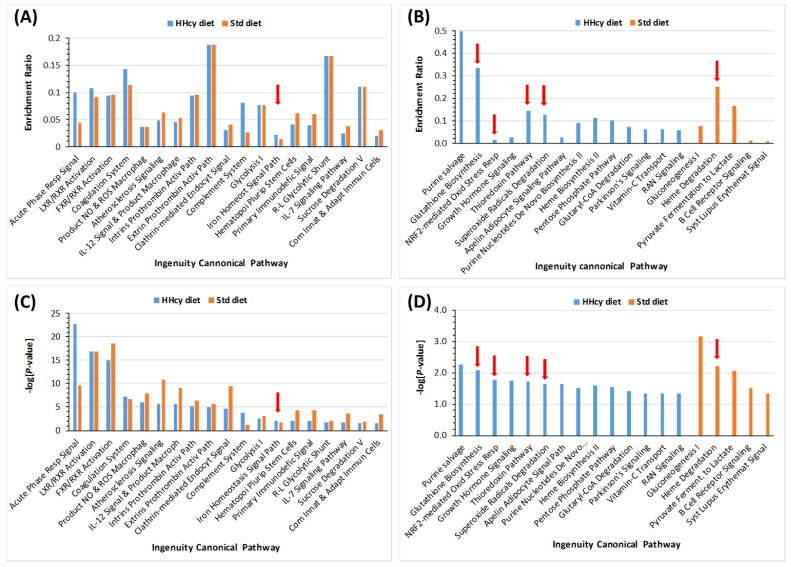
Enrichment ratios (**A**,**B**) and *p*-values (**C**,**D**) for indicated canonical pathways, identified by IPA, containing proteins affected by the *Pon1*^−/−^ genotype in mice fed the HHcy diet or a control diet. Benjamini–Hochberg, Benferroni, and false discovery rate corrections were applied to minimize the number of false positives. Red arrows highlight redox-related canonical pathways dysregulated by Pon1 depletion. Adapter from ref. [70].

**Figure 8 ijms-24-07764-f008:**
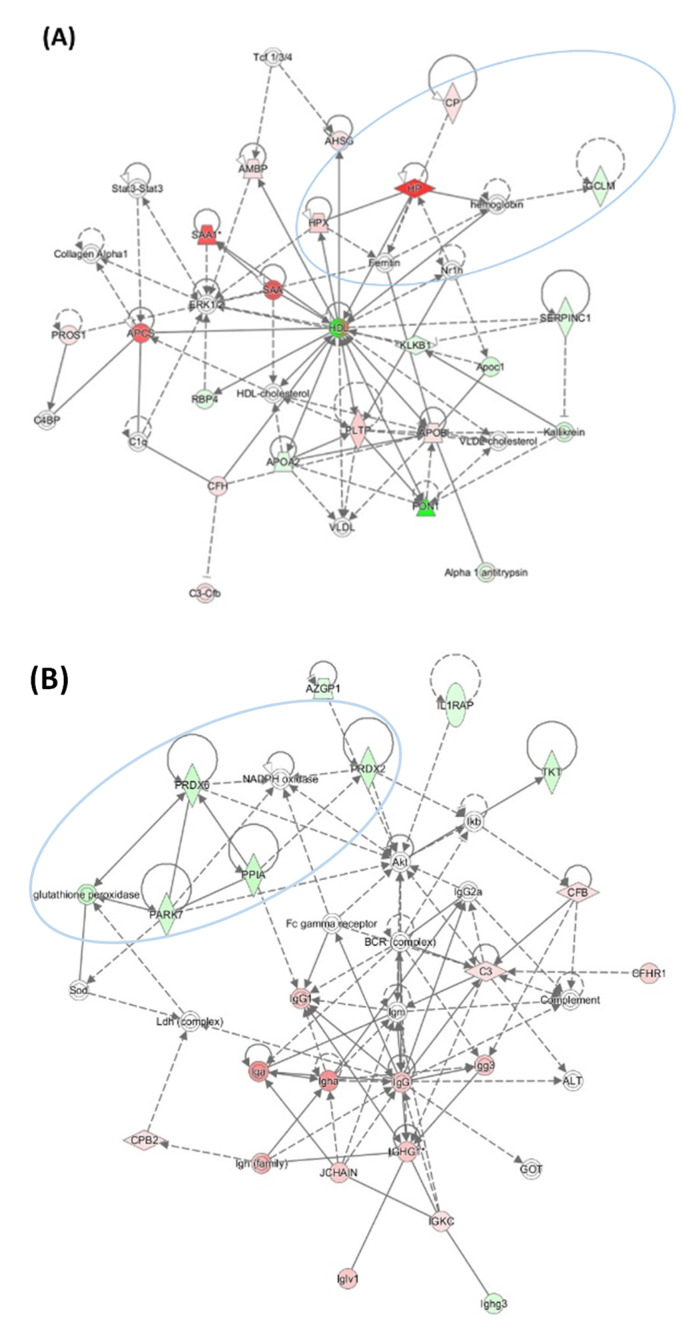
Top molecular networks associated with Pon1 depletion in mice fed with a HHcy diet. (**A**) Cardiovascular disease, organismal injury and abnormalities, and protein synthesis. This network contains proteins participating in iron metabolism/oxidative stress response (Cp, HP, HPX, and CGLM). (**B**) Hematological disease, humoral immune response, and inflammatory response. This network contains oxidative stress response proteins (Prdx2, Prdx6, Park7, and Ppia). (**C**) Cancer, cellular compromise, and inflammatory response. This network contains redox-related proteins (Cat, Ctsb, Grn, Gsn, Igfbp3, Pebp1, Serpina3, and Txn). Blue ovals highlight redox-related proteins dysregulated by Pon1 depletion. Adapted from ref. [70].

**Figure 9 ijms-24-07764-f009:**
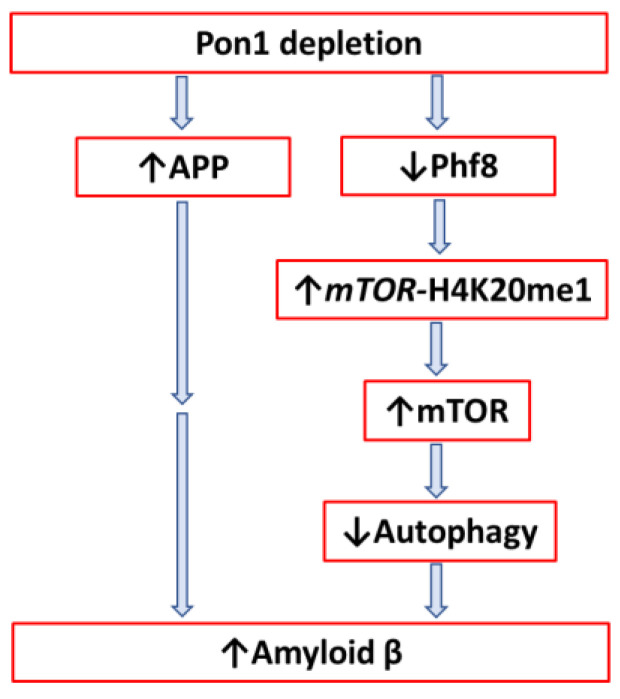
Schematic representation of the involvement of Pon1 depletion in amyloid β (Aβ) accumulation.

**Table 1 ijms-24-07764-t001:** Plasma proteins affected by the genetic depletion of PON1 in humans and mice *.

Unique to Mice (n = 41)	Unique to Humans (n = 12)	Proteins Affected Both in Mice and Humans (n = 9) ^#^
Oxidative stress ^†^ (n = 4): ↓Alb ^$^, ↓Blvrb, ↑α-1-microglubulin (Ambp), ↑Hemopexin (Hpx)	Oxidative stress ^†^ (n = 1): ↑Glutathione peroxidase 3 (GPX3)	Oxidative stress ^†^ (n = 3): ↑APOD↑, ↑APOM ^‡^↑, ↑haptoglobin (HP)↓
Immune response (n = 18): ↑Igh (n = 9), ↑Igj, ↑Igk (n = 6), ↑Igl (n = 2)	Immune response (n = 4): ↑CFP, ↓N/A, ↑PGLYRP2, ↑V2-6 (IGL)	Immune response (n = 1): ↓IGHG3↑
Acute phase response (n = 5): ↑Ahsg, ↑Orm1, ↑Orm2, ↑Saa1, ↑Saa2	Acute phase response (n = 1): ↑Ttr	Acute phase response (n = 1): ↑Ambp
Complement/coagulation (n = 7): ↑Al182371, ↑Cfh, ↑Clu ^$^, ↑F2 (prothrombin), ↓Klkb1, ↓Mbl1; ↓Serpinc1 (antithrombin III)	Complement/coagulation (n = 2): ↑C9, ↑V2-17 (IGL)	
Blood coagulation (n = 2): ↑Hrg ^‡^, ↓Itih1	Blood coagulation (n = 3): ↓PLG, ↓SERPINA10, ↓VTN	Blood coagulation (n = 1): ↓F13B↓
Lipoprotein/lipid metabolism (n = 5): ↓ApoA2, ↓ApoC2, ↓Azgp1, ↓Pgp, ↑Pltp		Lipoprotein metabolism (n = 4): ↓ApoA1, ↑ApoB, ↓ApoC1, ↓Pon1
Protein turnover (n = 1): ↑Mug1		Protein turnover (n = 1): ↑FETUB↓
Other (n = 6): ↓Afm, ↓Aldoa, ↓Bpgm, ↓Ica, ↓Ldha, ↓Lifr	Other (n = 1): ↓RBP4	

* Reanalyzed data from ref. [64] to highlight redox proteins affected by PON1 genotype/activity. Up and down arrows indicate the direction of change in protein levels. ^#^ Arrows left and right to the protein acronym refer to the change in protein levels in mice and humans, respectively. ^†^ The oxidative stress category was not highlighted in ref. [64]. ^$^ Enriched in, or ^‡^ largely excluded from, PON1-containing HDL subfraction of normal human HDL.

**Table 2 ijms-24-07764-t002:** Plasma proteins affected by in Pon1-depleted mice fed with HHcy or a control diet *.

Unique to HHcy Diet Mice (n = 66)	Unique to Control Diet Mice (n = 27)	Proteins Affected Both in HHcy and Control Diet Mice (n = 23) ^#^
Oxidative stress (n = 15): ↓Alad, ↑Cp, ↓Gclm, ↓Cat, ↑Ctsb, ↓Gsn, ↑Grn, ↓Prdx2 ^#^, ↓Prdx6, ↓Txn, ↓Igfbp3, ↓Park7 ^#^, ↓Pebp1 ^#^, ↓Ppia, ↓Serpina3k	Oxidative stress (n = 1): ↓Blvrb	Oxidative stress (n = 3): ↓Alb ^$^, ↑Hp, ↑Hpx
Immune response (n = 15): ↓Il1rap, Igh (n = 10↑, 1↓), ↑Igk (n = 3),	Immune response (n = 10): ↑Clu ^$^, ↑Igh (n = 3↑, 1↓), ↑Igk (n = 3), ↑Igl, ↑Igm	Immune response (n = 9): ↑Igh (n = 4), ↑Igj, ↑Igk (n = 3), ↑Igl
Acute phase response (n = 5): ↑Ahsg, ↑Orm1, ↑Orm2, ↑Saa1, ↑Saa2	Acute phase response (n = 1): ↑Ttr	Acute phase response (n = 1): ↑Ambp
Complement/coagulation (n = 6): ↑A2m ^$^, ↑Apcs, ↓F13a1, ↑C3, ↑Cfb, ↑Cfhr1	Complement/coagulation (n = 4): ↑AI182371, ↑F2, ↓F13b, ↓Mbl1	Complement/coagulation (n = 3): ↑Cfh, ↓Klkb1, ↓Serpinc1
Blood coagulation (n = 6): ↑Serpina10, ↓Gp1ba, ↑Gp5, ↑Itih3, ↑Pros1 ^$^, ↓Proz	Blood coagulation (n = 3): ↓Hgfac, ↑Hrg ^‡^, ↓Itih1	
Lipoprotein/lipid metabolism (n = 5): ↓ApoA2, ↓ApoC2, ↓Azgp1, ↓Pgp, ↑Pltp	Lipoprotein metabolism (n = 4): ↓Afm, ↑ApoD, ↑ApoM, ↑Lcat	Lipoprotein metabolism (n = 4): ↓ApoA1, ↑ApoB, ↓ApoC1, ↓Pon1
Protein turnover (n = 5): ↓Apeh, ↓Mug2, ↓Serpina3m, ↓Uba1, ↓Uba52	Protein turnover (n = 1): ↑Fetub	Protein turnover (n = 1): ↓Mug1;
Other proteins (n = 8): ↓Atic, ↓Nme1, ↓Pnp (purine metabo-lism), ↓Tpi, ↓Tkt (glucose metabolism), ↓Ran ^#^ (nucleoplasmic transport), ↓Rbp4 (retinol transport), ↓Spp2 (bone remodeling)	Other proteins (n = 3): ↓Aldoa, ↓Ldha (glucose metabolism), ↓Lifr (tissue regeneration)	Other proteins (n = 2): ↓Bpgm (glucose metabolism), ↓Ica (carbonic anhydrase inhibitor)

* Up and down arrows indicate the direction of change in protein levels. ^#^ Proteins also affected in the brain. ^$^ Enriched in, or ^‡^ largely excluded from, PON1-containing HDL subfraction of normal human HDL [5]. Adapted from ref. [70].

**Table 3 ijms-24-07764-t003:** Baseline PON1 arylesterase activity predicts cognition at the end of the study two years later—placebo group *.

Variable (n = 82–112)	Global Cognition				Episodic Memory				Attention/Processing Speed					
	MMSE_2 ^1^		TICSm_2 ^2^		HVLT-TR_2 ^3^		HVLT-DR_2 ^4^		Trail_Making_A _2 ^5^		SDMT_2 ^6^		SDMT_2 ^7^	
	β	*p*	β	*p*	β	*p*	β	*p*			β	*p*	β	*p*
Arylesterase activity_1	−0.24	0.034	−0.24	0.027	−0.19	0.046	−0.32	0.012	0.24	0.015	−0.18	0.008		
Paraoxonase activity_1		NS ^#^		NS ^#^		NS ^#^		NS ^#^					−0.33	0.028
*PON1-Q192R*		NS		NS		NS		NS		0.049			0.29	0.047
Brain atrophy rate	−0.27	0.029	−0.27	0.011		NS		NS		0.007	−0.24	0.001	−0.23	0.002
MMSE_1	0.26	0.017												
TICS-m_1			0.25	0.017										
HVLT-TR_1					0.45	0.000								
HVLT-DR_1							0.46	0.000						
Trail Making A_1									0.32	0.001				
SDMT_1											0.78	0.000	0.66	0.000
* Log-transformed data were used in analyses. _1—baseline _2—end of study	*p* = 0.000, R^2^ = 0.43		*p* = 0.000, R^2^ = 0.51		*p* = 0.000, R^2^ = 0.54		*p* = 0.001, R^2^ = 0.37		*p* = 0.000, R^2^ = 0.57		*p* = 0.000, R^2^ = 0.78		*p* = 0.000, R^2^ = 0.78	
	^1−7^ Adjusted for sex, age. Additional adjustment for: ^1, 4^ Anti-*N*-Hcy, tHcy_1; ^1^ *BDNF V66M* genotype; ^3^ Creatinine, *TCN 776CG* genotype; ^4^ *APOE* genotype; ^5^ Fe_1, FA_1, TG_1, *COMT V158M* and *DHFR 19bpins* genotypes. ^#^ Models with or w/o arylesterase. MMSE—Mini-Mental State Examination; TICS-m—Telephone Inventory for Cognitive Status modified; HVLT-TR_1—Hopkins Verbal Learning Test-revised Total Recall; HVLT-DR—Hopkins Verbal Learning Test-revised, Delayed Recall; SDMT—Symbol Digits Modalities Test.													

**Table 4 ijms-24-07764-t004:** B vitamin therapy prevents the association of baseline PON1 arylesterase activity with cognition at the end of the study—B vitamin group *.

Variable (n = 82–112)	Global Cognition				Episodic Memory				Attention/Processing Speed					
	MMSE_2 ^1^		TICSm_2 ^2^		HVLT-TR_2 ^3^		HVLT-DR_2 ^4^		Trail_Making_A _2 ^5^		SDMT_2 ^6^		SDMT_2 ^7^	
	β	*p*	β	*p*	β	*p*	β	*p*			β	*p*	β	*p*
Arylesterase activity_1		NS		NS		NS		NS		NS		NS		
Paraoxonase activity_1														NS
*PON1-Q192R*										NS				NS
Brain atrophy rate		NS		NS		NS		NS		NS		NS		NS
MMSE_1	0.52	0.000												
TICS-m_1			0.38	0.002										
HVLT-TR_1					0.63	0.000								
HVLT-DR_1							0.47	0.001						
Trail Making_1									0.62	0.000				
SDMT_1											0.65	0.000	0.66	0.000
* Log-transformed data were used in analyses. _1—baseline _2—end of study	*p* = 0.019, R^2^ = 0.20		*p* = 0.001, R^2^ = 0.28		*P* = 0.000, R^2^ = 0.38		*p* = 0.001, R^2^ = 0.20		*p* = 0.005, R^2^ = 0.30		*P* = 0.000, R^2^ = 0.61		*p* = 0.000, R^2^ = 0.60	
	^1−7^ Adjusted for sex, age. Additional adjustment for: ^1, 4^ Anti-*N*-Hcy, tHcy_1; ^1^ *BDNF V66M* genotype; ^3^ Creatinine, *TCN 776CG* genotype; ^4^ *APOE* genotype; ^5^ Fe_1, FA_1, TG_1, *COMT V158M* and *DHFR 19bpins* genotypes. Neuropsychological test acronyms defined as in Table 3.													

**Table 5 ijms-24-07764-t005:** Brain proteins affected by Pon1 depletion and/or HHcy in mice are also affected in AD and other neuropathies.

Protein Name	Change in *Pon1^−/−^* vs. *Pon1^+/+^* Brain *		Change in 1% Met Diet vs. Std. Diet Brain *	Change in AD Brain (Other Neuropathy or Animal Model) **
	Std. Diet	1%-Met Diet	*Pon1^+/+^*	
Brain-specific				
Ncald	–	**↑**	**↓**	**↓**, (**↓** in *Gls*^−/−^ mouse)
Nrgn	**↓**	**↑**	**↓**	**↓**
Stmn1	–	**↑**	**↓**	**↓**, (**↑** in MS, TLE, SMA, schizophrenia), (**↑** in HD4 mouse model)
Antioxidant defense				
Sod1	**↓**	–	–	(**↑** in ALS)
Prdx2	–	**↑**	**↓**	**↑**
DJ-1 (Park7)	**↓**	**↑**	**↓**	**↑**
Energy metabolism				
Ak1	–	**↑**	**↓**	**↑**
Cell cycle				
GDI1	–	**↑**	**↓**	(**↑** in rat ischemic brain)
Ran	–	**↑**	**↓**	**↑**
Cytoskeleton assembly				
Tbcb	**↓**	**↑**	**↑**	(**↑** in GAN)
CapZa2	**↑**	–	**↑**	**↑** CapZb2 ^#^
Other proteins				
Hdhd2	–	**↑**	–	

* The up “**↑**” and down “**↓**” arrows indicate up-regulated and down-regulated proteins, respectively. The dash (—) indicates no significant change. ** AD—Alzheimer’s disease; GAN—giant axon neuropathy; HD—Huntington disease; MS—multiple sclerosis; PD—Parkinson’s disease; SMA—spinal muscular atrophy; TLE—temporal lobe epilepsy. ^#^ The b2 subunit of the CapZ heterodimer. Adapted from ref. [98].

## Data Availability

The data that support the findings of this study are available in the methods of this article.

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
