# Peer review of "Proteomic Exploration of Paraoxonase 1 Function in Health and Disease"

_ijms, 2023, doi:10.3390/ijms24097764_

Round 1

Reviewer 1 Report

The paper of Hieronim Jakubowski "Proteomic Exploration of Paraoxonase 1 Function in Health and Disease" is a review article on a family of enzymes, that are my favourite ones.

The manuscript is decently written and organized with the strong message to the reader. Before publishing in IJMS there are a few more things that needs to be addressed first.

Comments and Suggestions for Authors

1. Please explain “The PON1 activity with Hcy-thiolactone as a substrate (HTLase) (Figure 1) is strongly correlated with the activity with paraoxon in various…”. It is not clear what kind of correlation is it (positive, negative, strong, weak….)

2. Discuss what is specificity of PON1 regarding to Hcy-thiolactone in comparison with other artificial substrates.

3. Discuss about other serum esterases and their effects on Hcy-thiolactone.

4. What are the limitations of measuring HTLase activity in serum?

5. Discuss about HTLase activity in CSF?

6. Explain “Hcy-thiolactone levels are significantly higher in carriers of the low activity PON1-192QQ alleles compared with carriers of the high activity PON1-192RR alleles”. It is not clear what is low activity PON1-192QQ alleles and high activity PON1-192RR alleles? Activity to what, DHC, PA, PX….

7. Discuss how arylesterase activity reflects the concentration of PON1 (ref. 5, 24). If specific enzyme activity is measured, then it is not possible to determine that enzyme activity correlates with concentration. Only kinetic parameter Vmax which is determined from initial velocity approach or progress curves reflects PON1 concentration.

8. Discuss about novel kinetic approach, which has been used recently for determination of PON1 kinetic parameters (determination of Km and Vmax directly from individual time-concentration progress curves) and compare it with classical enzyme specific activity (PON1 activity) approach in a clinical context, especially to our understanding of neurodegenerative and CV diseases. Advantages and disadvantages.

9. The sentence “the 192Q allele is associated with low paraoxonase and high arylesterase activity while the 192R allele associates with high paraoxonase and low arylesterase activity” is not entirely true. Indeed, kinetic parameter Km for phenyl acetate on purified serum PON1 192Q is lower in comparison to Km for PON1 192R isoform, however, the ratio Vmax/Km which defines the catalytic activity, is for both isoforms the same (Billecke et al. 2000).

10. Reference is missing for the following text “Relationships between PON1, lipid peroxidation and dementia were examined pa-tients with AD (n = 63), vascular dementia (n = 40) and mixed dementia (n = 33). Malondialdehyde/thiobarbituric acid reactive substances were increased more in vascular dementia than in AD. In patients with vascular involvement, increase in MDA/TBARS reflected the extent of global cortical atrophy. PON1 arylesterase activity was decreased in patients with dementia, more so in patients with severe cognitive deficits. In patients with vascular dementia, a reduction in PON1 arylesterase activity reflected the increase in medial temporal lobe atrophy and brain ischemia. The authors concluded that PON1 activity and MDA/TBARS marker of oxidative stress were associated with vascular de-mentia and brain atrophy rather than cognitive decline.«

Reviewer decision

The article is extremely interesting, well written and will have high impact within PON1 community. I suggest publishing in IJMS after author critically discuss issues addressed by the reviewer.

Author Response

We thank Reviewer 1 for her/his constructive comments that helped to improve the manuscript. My responses to the Reviewer's comments and, where applicable, corresponding changes in the manuscript (highlighted in yellow) are described below.

  1. In response to the Reviewer’s comment, a new Figure 2 has been added and the questioned sentence has been clarified to read "…strongly positively correlated with the paraoxonase activity in various populations (Figure 2A, B)...". To provide an illustrative support for the statement that Hcy-thiolactone is a naturally occurring substrate of PON1, new Figure 3 has been added.
  2. As described in the introductory section of the manuscript, PON1 hydrolyzes thioesters, esters, and phosphoesters, which indicates that PON1 is a nonspecific hydrolytic enzyme.
  3. To clarify this point, the following sentences has been included "PON1 is responsible for essentially all HTLase activity in the plasma [11, 12] while BLMH [13] is a major HTLase in other tissues [15]."
  4. The limitation of measuring HTLase activity is that it requires a radiolabelled 35S-Hcy-thiolactone.
  5. There is no information in the published papers regarding HTLase activity in the CSF.
  6. To clarify this point, the questioned sentence has been modified to read " Hcy-thiolactone levels are significantly higher in individuals with low HTLase/paraoxonase activity (PON1-192QQ) compared with individuals with high HTLase/paraoxonase activity (PON1-192RR) [23], suggesting that the paraoxonase activity might reflect the physiological HTLase activity of PON1 (Figure 3A).
  7. This statement is supported by quantification of the PON1 protein using an anti-PON1 antibody, as now indicated in the text: “The arylesterase activity is much less affected by the PON1-Q192R variation than the paraoxonase activity [5, 23, 24, 38, 39] and appears to reflect the concentration of the PON1 protein [5, 24]. Although Hcy-thiolactone levels in humans are affected by the paraoxonase activity (Figure 3A), they are not affected by the arylesterase activity (Figure 3B) or PON1 protein levels (Figure 3C) measured with an anti-PON1 antibody [23].”
  8. We appreciate the Reviewer's suggestion. However, a discussion of basic enzymological assays would not fit the scope of this manuscript. 
  9. This statement is based on assays performed at least in dozen or so labs.  Activity refers here to the rate, not the catalytic efficiency, of the enzymatic reaction. 
  10. Thank you for catching this omission. The missing reference has now been included in the manuscript.

Reviewer 2 Report

A very interesting study, there are many recent articles on PON1 finding new uses in different pathologies. The current article provides a very interesting insight.

I'm not the most experienced in interpreting Top molecular networks.

Always write PON1 in the same way, in capital letters, review it.

Adapt the title to the content of the article, rather than refer to specific pathologies.

It remains to explain the methodology, how the reviewed studies have been selected, etc.

Author Response

I thank Reviewer 2 for her/his positive comments.

Comment: write PON1 in the same way, in capital letters, review it.

Response: according to the accepted nomenclature rules, PON1 refers to the human protein, while Pon1 refers to the rodent protein.

Comment: Adapt the title to the content of the article, rather than refer to specific pathologies.

Response: The manuscript's title does nor refer to specific pathologies.

Comments: It remains to explain the methodology, how the reviewed studies have been selected, etc. 

Response: the review includes studies identified by searching medline against terms "PON1/paraoxonase 1/homocysteine thiolactonase and cardiovascular diesease", "PON1/paraoxonase 1/homocysteine thiolactonase and Alzheimer's disease"

Reviewer 3 Report

Dear author,

I have been delighted to read your paper because the field of Paraoxonase 1 has immense interest to me. 

I have learned some important insights about the lactonase activity of PON1 because prior to reading your paper I thought the same as you stated it could be awkward to say that the native activity of the enzyme is the lactonase activity. Based on in vitro studies, it indeed is difficult to state that this is a universal truth.

My only observation related to this is that the title of Figure 1 should avoid the ”Only” because this could imply that you have evaluated all types of lactones and only Hcy-thiolactone was found as a natural substrate.

I like also the point of view regarding the redox activity of PON1, which is currently largely over-exaggerated. The cited references are quite on-point.

The review is an important catalyst for further studies in the field of PON1 research. I will advocate for this paper to be published in this prestigious journal.

Good luck!

Author Response

I thank Reviewer 3 for her/his positive comments.

Regarding a comment on the title of Figure 1: I have followed the Reviewer's suggestion and deleted the word "only".

Reviewer 4 Report

I have received Jakubowsky's manuscript in the second round of review. The author has satisfactorily answered the questions and comments of the rest of the reviewers. The article is exhaustive and very interesting. Some parts are provocative and controversial, but that is not a bad thing. On the contrary, it is positive to point out that some aspects taken for granted about PON1 may not be so much, and debate and controversy are always the motors of science. I will not add any more suggestions or criticisms, and I think the article should be published as is.

Round 2

Reviewer 1 Report

  Author has critically discussed all issues addressed by the reviewer, I suggest publishing in IJMS.

Reviewer 2 Report

A very interesting study, the current article provides a very interesting insight.